# Evaluation and Empirical Research on Eco-Efficiency of Financial Industry Based on Carbon Footprint in China

Xiaolan Chen [1], Kaikai Wang [2], Guanjiang Wan [3], Yufei Liu [2], Wenbin Liu [4], Wanfang Shen [5,*] and Jianing Shi [2]

1. Shandong Province Social Governance Intelligent Technology Innovation Center, Shandong University of Finance and Economics, Jinan 250014, China
2. School of Statistics and Mathematics, Shandong University of Finance and Economics, Jinan 250014, China
3. Kent Business School, University of Kent, Canterbury CT2 7NZ, UK
4. Faculty of Business and Management, Beijing Normal University—Hong Kong Baptist University United International College, Zhuhai 519087, China
5. Shandong Key Laboratory of Blockchain Finance, Shandong University of Finance and Economics, Jinan 250014, China
* Correspondence: wfshen@sdufe.edu.cn

**Abstract:** Since finance is the core of economic development, the green development of the financial industry is an essential driving force not only for achieving the dual "carbon" goal of China but also for economic and social sustainable development. An accurate understanding of the ecological efficiency of the financial industry is of great importance for guiding sustainable economic development. In this paper, we first calculate the carbon footprint of China's financial industry in 2012 and 2017 based on the life cycle theory and the input–output analysis method. Second, we analyze the primary sources and final flows of the carbon footprint of the financial industry in each province from the perspectives of the industrial chain and final demand. Finally, we estimate the ecological efficiency, emission reduction, and value-added potential of the financial industry by using the radially adjusted slack variable DEA model (SRAM-DEA) under two assumptions, natural disposability and managerial disposability. The results show that (1) the ecological efficiency of the financial industry in most provinces is low, and the regional differences are significant; (2) the overall ecological efficiency of the financial industry in 2017 was better than that in 2012; (3) technological innovation of financial products and the upgrading of capital supervision play an essential role in promoting the improvement of ecological efficiency. Especially, under managerial disposability, the ecological efficiency of the financial industry in each province has a greater potential for emission reduction and added value.

**Keywords:** green finance; carbon footprint; ecological efficiency; SRAM-DEA model

## 1. Introduction

Since China has become the second-largest economy in the world, the overall strength and competitiveness of the country have improved significantly. However, the rapid development of the economy has also produced serious environmental pollution problems. In order to solve these problems, the government of China (GOC) has taken active measures. Specifically, to achieve the goal of balanced development of the environment, economy, and society, the GOC actively introduced environmental protection policies and regulations and guided funds to flow from 'three-highs' enterprises to green low-pollution enterprises. As the main industry that supports financial development closely related to carbon emissions, the financial industry plays an increasingly important leading role in the promotion of green economic development [1–3]. The green development of the financial industry is an important driving force for promoting economic and socially sustainable development. Green finance is gradually incorporated into the national top-level design. As the core industry of economic development, the financial industry has shown a steady increase

in added value in the past 20 years (see Figure 1). It has always been running upstream, midstream, and downstream of the industrial chain of various industries. We cannot ignore the impact of the financial industry on environmental quality. Measuring the impact of the financial industry on the ecological environment is of great significance for guiding the green development of the financial industry.

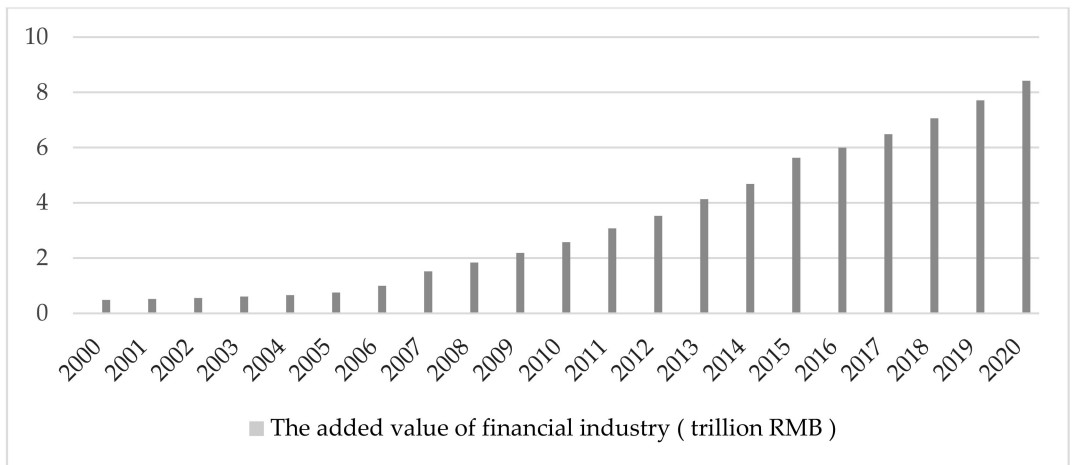

**Figure 1.** The added value of China's financial industry in the past 20 years.

There are many studies on the effects of the development of the financial industry on the ecological environment [4,5], but there is a lack of research on the efficiency of measuring the impact of the financial industry on the environment. Existing research focuses on the impact of green financial development or financial agglomeration on regional ecological efficiency. Such studies regard the financial industry as an important factor affecting the quality of the regional ecological environment. At the same time, this kind of research shows that the financial industry impacts regional ecological efficiency mainly through capital support, capital allocation, and financial supervision [4,5]. Unbalanced capital allocation, lack of supervision, and wrong capital flow will have a negative impact on environmental quality. This is not the direct impact of the financial industry on the ecological environment, but the indirect impact of the formation of the ecological environment through the mobilization and supervision of capital, which provides a new perspective for the study of financial and ecological efficiency. However, the financial industry belongs to the tertiary industry and has a service output. Compared with energy-consuming industries, its direct energy consumption is minimal, and general pollutant emissions caused directly cannot be measured. This has led to few studies that directly measure its own ecological efficiency, and there is no authoritative definition of the ecological efficiency of the financial industry. Therefore, the main purpose of this paper is to explore the method of calculating the ecological efficiency of the financial industry, and to measure the ecological efficiency of the financial industry in various provinces of China.

Schaltegger and Sturm introduced the concept of eco-efficiency in 1990, that is, the ratio of added value to increased environmental impacts, which has since been widely recognized and accepted. It was first cited in 1992 by the World Business Council for Sustainable Development (WBCSD) in the book 'Changing course: a global business perspective on development and the environment.' In recent years, ecological efficiency has been gradually used to analyze products, enterprises, industries, and even regions [6]. Theoretical research on ecological efficiency mainly focuses on two aspects. First, the regression model is used to explore the main factors influencing environmental efficiency. For example, Yu et al. [7] studied the regional differences, the dynamics of distribution, and the convergence of ecological efficiency in China's urban agglomerations. Li et al. [8] calculated the eco-efficiency of industrial land in China. They tested the influence mechanism of variables on the eco-efficiency of industrial land in different regions by the Tobit

model. Zeng explored regional differences and influencing factors of ecological efficiency in China based on spatial panel data [9]. Second, the Data Envelopment Analysis (DEA) model is used to study environmental efficiencies. For example, Kang Lei et al. [10] used the DEA method to explore the resource productivity and ecological efficiency of various provinces, autonomous regions, and municipalities in China, and the resource productivity and environmental efficiency under a two-stage system. Chi et al. [11] used the SBM model, combined with the traditional spatial Markov probability transfer matrix, to calculate the provincial agricultural ecological efficiency in China. Such studies will select different indicators for measuring eco-efficiency based on research objects. Considering the particularity that the direct energy consumption of the financial industry and the general pollutant emissions cannot be measured, it is necessary to find a combination point between the development of the financial industry and the performance of the ecological environment. Therefore, this paper takes the 'carbon footprint' based on the life cycle theory as an essential combination point, and uses the life cycle theory and input–output analysis method to reflect the carbon footprint of the financial industry through capital flows.

As an essential indicator of ecological efficiency, the term 'carbon footprint' is usually used to describe direct and indirect carbon emissions of products over their life cycles. With the continuous exploration of sustainable economic development, the carbon footprint has gradually become an important indicator for measuring the high-quality development of the regional economy. It has been applied to the analysis of sustainable development in all industries. For example, Wang et al. [12] proposed two methods of investigating carbon footprint: the input–output method and the process analysis method. Hu et al. [13] used the carbon footprint method to measure the carbon emissions of the logistics industry in the Yangtze River Economic Belt. They applied this method to the logistics and supply chain management decisions. Wang et al. [14] calculated the carbon emissions of the Chinese wood-based panel industry based on international carbon footprint standards, and then studied the carbon emission reduction path. Li et al. [15] used the life cycle theory to evaluate the carbon footprint of the delivery of Chinese construction. At the same time, carbon footprint research based on life cycle theory shared important methodological significance by studying various carbon emissions. For instance, Zhou et al. [16] applied the life cycle theory to calculate the carbon footprint of China's construction industry. The existing authoritative collection of carbon emissions data is mainly based on the direct carbon emissions of the primary and secondary industries. Carbon emissions from the tertiary industry are difficult to measure directly. Research on ecological efficiency and carbon emissions focuses mainly on regional studies or energy-consuming industries. For example, Wang et al. [17] evaluated and analyzed the industrial carbon emission efficiency in the Yangtze River Basin; Yin et al. [18] conducted a spatial and regional comparative analysis of carbon emissions. However, there is a lack of appropriate methods to measure the carbon emissions of the financial industry, and there is little literature studying the carbon emissions and ecological efficiency of the financial industry.

Based on the analysis presented above, the core purpose of this paper is to measure the ecological efficiency of the financial industry. Considering the production process of the financial industry, this paper defines the eco-efficiency of the financial industry as the ratio of the investment required for the development of the financial industry to the environmental impact. The efficiency value reflects the ecological environment performance in the development of the financial industry. In this paper, based on life cycle theory and input–output analysis, we first calculate the carbon footprint of the financial industry in various provinces of China in 2012 and 2017. Second, from the perspective of the industrial chain and final demand, we analyze the primary source and final flow of the carbon footprint of the financial industry in each province. Third, we use the carbon footprint as an important undesirable output in the environmental index system to evaluate the ecological efficiency of the financial industry using the DEA model. Specifically, the radially adjusted slack variable DEA model (SRAM-DEA) is used to calculate the ecoefficiency, the potential for emission reduction, and value-added of the financial industry in each

province in 2012 and 2017 under the conditions of natural disposability and managerial disposability, respectively.

The rest of this paper is as follows: Section 2 introduces the construction of an index system, carbon emission calculation model, and evaluation model. Section 3 measures and analyzes the carbon emission intensity of the industry, the carbon footprint of the financial industry in each province, and the ecological efficiency of the financial industry in each province. Section 4 summarizes the paper and puts forward policy recommendations to improve the ecological efficiency of the financial industry.

## 2. Ecological Efficiency Evaluation Index System and Model Construction

### 2.1. Index Selection

Compared with the primary industry and the secondary industry, the financial industry belongs to the tertiary industry, providing financial services to companies or project operations. Its special industrial nature and service characteristics make it impossible to measure energy input and direct pollutant emissions. Therefore, it is necessary to measure its ecological environment output through indirect indicators. This paper uses the carbon footprint in its life cycle to characterize the important indicators of the ecological environment factors of the financial industry. At the same time, the development of the financial industry needs investment. Based on the Cobb–Douglas production function and the operation of the financial sector, the number of employees in the financial industry at the end of the year is chosen as the labor input index. The investment in fixed assets is selected as the capital input index. The value added by the financial industry at the end of the year is chosen as the desirable output index [19]. Since the carbon footprint is the crucial index of environmental impact, it is used as the undesirable output of the ecological efficiency evaluation to measure carbon emissions in the life cycle of the financial industry. The ecological efficiency evaluation index system of the financial industry is shown in Table 1.

**Table 1.** Eco-efficiency evaluation index system of the financial industry.

| | | |
|---|---|---|
| Input | Fixed asset investment | |
| | Employed persons | |
| Output | Undesirable output | Carbon footprint |
| | Desirable output | Value-added |

### 2.2. Construction of Carbon Footprint Measurement Model

Since 'double carbon' was put forward in China, the carbon emission database has been gradually improved, and the carbon emission measurement technology has steadily matured. However, the existing authoritative database focuses on regional carbon emissions, or carbon emissions of the energy-consuming industries, but there is no detailed calculation for the service industry. There is no direct energy consumption or environmental pollution in the financial industry, and the financial industry and ecological environment benefits are reflected in the financial services. Therefore, it is more meaningful to examine the carbon footprint in the life cycle of the financial industry than to measure the carbon emissions of the financial industry itself.

Life cycle theory refers to the process of an organism from generation to extinction, considering all stages of the existence of life. With the development of economies, life cycle theory has been applied to enterprises and products, and enterprise life cycle theory and product life cycle theory have been developed. The life cycle of the financial industry includes its development, service generation, service process, and service end. The calculation of the carbon footprint of the financial industry revolves around these parts. In previous studies, the life cycle theory often appears simultaneously with the input–output model.

Based on the good mathematical properties of the input–output model proposed by American economist W. Leontief [20], the interdependent economic relationship between various industries in the financial system is applied to the transformation of the monetary flow relationship between industries into the emission flow relationship. The relationship between direct and indirect carbon emissions is determined clearly [21], which is soon used to study environmental and energy issues [22,23]. It is very applicable to apply the input–output model to the carbon footprint in the life cycle of the financial industry. However, China's input–output table is updated every five years. So far, the input–output table has only been published until 2017. Thus, in this paper, we chose data from 2012 and 2017.

The carbon footprint measurement model of the financial industry including both intermediate inputs and the final demand is as follows:

Basic equilibrium relation based on the input–output model:

Intermediate demand + Final demand-inflow = Total output

Then we can obtain output equilibrium equations

$$\sum_{j=1}^{n} x_{ij} + y_i = x_i (i = 1, 2, \cdots, n) \tag{1}$$

where $x_i (i = 1, 2, \cdots, n)$ represents the total output level of the industry, $x_{ii}$ represents the medium demand used by the industry itself, $x_{ij} (j = 1, 2, \cdots, n)$ represents the intermediate usage provided by industry $i$ for industry $j$, and $y_i (i = 1, 2, \cdots, n)$ represents the final demand of the industry $i$, including household consumption, government consumption, exports, and reserves.

By introducing the direct consumption coefficient $a_{ij} (i, j = 1, 2, \cdots, n)$, we can obtain the value of industry $i$ that is consumed by industry $j$ to produce one unit of value as follows:

$$a_{ij} = \frac{x_{ij}}{x_j} (i, j = 1, 2, \cdots, n) \tag{2}$$

From Equations (1) and (2), we have

$$\begin{cases} a_{11}x_1 + a_{12}x_2 + \cdots + a_{1n}x_n + y_1 = x_1, \\ a_{21}x_1 + a_{22}x_2 + \cdots + a_{2n}x_n + y_2 = x_2, \\ \qquad \cdots \\ a_{n1}x_1 + a_{n2}x_2 + \cdots + a_{nn}x_n + y_n = x_n. \end{cases} \tag{3}$$

Then let $X = (x_1, x_2, \cdots, x_n)'$, $Y = (y_1, y_2, \cdots, y_n)'$, $A = (a_{ij})_{n \times n}$, we can express formula (3) as follows

$$AX + Y = X \text{ or } (E - A)X = Y \tag{4}$$

Then we can obtain

$(E - A)^{-1}Y = X$, where $(E - A)^{-1}$ is the inverse matrix of $(E - A)$.

We suppose that the input–output tables compiled by provinces, intermediate inputs, and imports in final demand are not distinguished [24]. Therefore, the import volume is proportionally allocated to intermediate inputs and final demand, and the direct consumption matrix and final demand are obtained as follows.

$$A^1 = A(\frac{X}{X + IM}), Y^1 = Y(\frac{X}{X + IM}).$$

where $IM$ represents the import volume.

In conclusion, we obtain the input–output equilibrium model

$$X = \left(E - A^1\right)^{-1} Y^1 \tag{5}$$

According to the product of various energy consumption and various energy carbon emission factors of each industry, the carbon emissions of each industry are calculated. The

ratio of carbon emissions of the industry to the output value of the industry is defined as the direct carbon emission intensity coefficient of the industry [25]:

$$p_i = \frac{\sum_{k} energy_{ik} \times energytype_k}{output_i}, (i = 1, 2, \cdots, n; k = 1, 2, \cdots, m) \tag{6}$$

where $p_i$ represents the direct carbon emission intensity factor per unit output of industry $i$; $energy_{ik}$ represents the number of energy $k$ consumed by industry $i$; $energytype_k$ represents the energy $k$ carbon emission factor, which is the product of energy 9 converted into standard coal coefficient and energy carbon emission coefficient, and $output_i$ represents the output value of industry $i$.

Based on Equations (5) and (6), the carbon emission calculation model can be expressed as:

$$C = P(E - A^1)^{-1}\overset{\wedge}{Y} \tag{7}$$

where $\overset{\wedge}{Y}$ is the diagonal matrix composed of the final demand.

*2.3. Construction of DEA Evaluation Model for Ecological Efficiency*

2.3.1. Production Possibility Sets Based on Natural and Managerial Disposability

Decision-Making Units (DMU) reduce undesirable outputs in two ways. One of the two ways is to reduce the input to reduce the undesirable output while increasing the desirable output as much as possible. This path is a production path under natural disposability. Of course, increasing the opportunity cost by reducing the operating cost will not change the negative impact of high pollution in decision-making units. Therefore, this path is a non-positive one for dealing with environmental regulation. The second is the production path under managerial disposability. The decision-making units reduce the undesirable output and increase the desirable output by increasing the input of technological innovation or management reform. In this way, the decision-making units increase the investment cost, but reduce the operation and opportunity cost. The desirable output can be increased while the undesirable output is reduced, so this is a positive path. Therefore, the production possibility set is constructed in these two ways.

Based on the characteristics of natural disposability and managerial disposability, Sueyoshi et al. [26] proposed the following Production Possibility Sets (PPS) with undesired outputs under this disposability.

$$P^N(X) = \{(Y, Z) | X \geq \sum_{j=1}^{n} \lambda_j X_j, Y \leq \sum_{j=1}^{n} \lambda_j Y_j, Z \geq \sum_{j=1}^{n} \lambda_j Z_j, \sum_{j=1}^{n} \lambda_j = 1, \lambda_j \geq 0\} \tag{1*}$$

$$P^M(X) = \{(Y, Z) | X \leq \sum_{j=1}^{n} \lambda_j X_j, Y \leq \sum_{j=1}^{n} \lambda_j Y_j, Z \geq \sum_{j=1}^{n} \lambda_j Z_j, \sum_{j=1}^{n} \lambda_j = 1, \lambda_j \geq 0\} \tag{2*}$$

where $X \in R_+^m$ represents the input vector with $m$ elements, $Y \in R_+^s$ represents the desirable output vector with $s$ elements, and $Z \in R_+^h$ represents the undesirable output vector with $h$ elements. The common goal of natural disposability and managerial disposability is to reduce the undesirable output and increase the desirable output, but the constraints of their input variables are opposite.

2.3.2. Construction of DEA Comprehensive Evaluation Model

The data envelopment analysis model (DEA) is widely used in efficiency measurement and evaluation. For example, Huang et al. [27] used the DEA model to measure the energy utilization efficiency of each province in China. Li al. [28] measured the land-use efficiency of 65 county-level cities in the Yellow River Basin using data envelopment analysis (DEA).

For different constraints and objectives, the traditional DEA model is constantly improved in the development process. Sueyoshi et al. [29] studied the environmental RAM model based on the traditional DEA model and realized the evaluation contained environmental factors. Li [30] used the RAM-DEA model to study the internal relationship between China's carbon emissions and economic growth. Chen et al. [31] used the RAM-DEA model to study the energy crowding impact on China's manufacturing industry. Meng et al. [32] analyzed the low-carbon economic efficiency of China's industrial sector based on the unexpected output RAM model. Sueyoshi et al. [26] first discussed the SRAM model and applied this model to evaluate the environmental effects of American thermal power plants. Based on the RAM model, we use the radially adjusted slack variable DEA (SRAM-DEA) model to study the ecological efficiency evaluation of the financial industry in 30 provinces of China in this paper.

According to the definition of axiomatic PPS (1*, 2*), the output-oriented SRAM-DEA model under natural disposability (EN) and managerial disposability (EM) is defined as follows:

$$Max \left( \theta + \varepsilon (\sum_{i=1}^{m} R_i^x d_i^x + R^y d^y + R^z d^z) \right)$$

$$s.t. \begin{cases} \sum_{j=1}^{n} x_{ij}\lambda_j + d_i^x = x_{i0}(EN), or, \sum_{j=1}^{n} x_{ij}\lambda_j - d_i^x = x_{i0}(EM)(i = 1, 2, \cdots, m), \\ \sum_{j=1}^{n} y_j\lambda_j - d^y - \theta y_0 = y_0, \\ \sum_{j=1}^{n} z_j\lambda_j + d^z + \theta z_0 = z_0, \\ \sum_{j=1}^{n} \lambda_j = 1, \\ \lambda_j \geq 0 (j = 1, 2, \cdots, n), d_i^x \geq 0 (i = 1, 2, \cdots, m), d^y \geq 0, d^z \geq 0. \end{cases}$$
(8)

where $d^x, d^y, d^z$ represent the slack variables of labor, capital input, value-added, and carbon emissions, respectively; $R$ is the range determined by the upper and lower bounds of the input and output, which are defined as:

$$R_i^x = (m + s + h)^{-1} (\max\{x_{ij} | j = 1, 2, \cdots, n\} - \min\{x_{ij} | j = 1, 2, \cdots, n\})^{-1},$$
$$R^y = (m + s + h)^{-1} (\max\{y_j | j = 1, 2, \cdots, n\} - \min\{y_j | j = 1, 2, \cdots, n\})^{-1},$$
$$R^z = (m + s + h)^{-1} (\max\{z_j | j = 1, 2, \cdots, n\} - \min\{z_j | j = 1, 2, \cdots, n\})^{-1} (i = 1, 2, \cdots, m).$$

According to the optimal efficiency score and slack variables obtained by model (8), the financial industry ecological efficiency score is expressed as:

$$EN = 1 - [\theta^* + (\sum_{i=1}^{m} R_i^x d_i^{x*} + R^y d^{y*} + R^z d^{z*})].$$
(9)

## 3. Empirical Study

### 3.1. Carbon Footprint Calculation

3.1.1. Industry Carbon Emission Intensity Coefficient

Since the input–output table is updated every five years, this paper studies the carbon footprint of the financial industry in 2012 and 2017 in 30 provinces using the 'China Regional Input–Output Table.' The data are derived from the official network of the National Bureau of Statistics, the official network of the China Input–Output Society, and the 2017 China Regional Input–Output Table [33]. The annual energy consumption data of various industries and the energy equivalent standard coal coefficient are derived from the China Energy Statistics Yearbook. The energy carbon emission coefficients other than electric energy are derived from the IPCC experimental data. Considering the vast amount of electricity used in production and life, this paper also incorporates electricity in calculating the intensity of industrial carbon emissions. The energy carbon emission

coefficient of electricity comes from the analysis of Chinese scholars [34]. The specific carbon emission coefficients of coal and energy are shown in Table 2.

**Table 2.** Energy conversion standard coal coefficient and energy carbon emission coefficient.

| Type of Energy Source | Energy Standard Coal Coefficient | Energy Carbon Emission Coefficient (t Coal/Standard Coal t) | Energy Carbon Emission Intensity Coefficient |
|---|---|---|---|
| Coal | 0.7143 (Tons of standard coal/tons) | 0.682 | 0.487 |
| Coke | 0.9714 (Tons of standard coal/tons) | 0.765 | 0.743 |
| Crude oil | 1.4286 (Tons of standard coal/tons) | 0.676 | 0.966 |
| Gasoline | 1.4714 (Tons of standard coal/tons) | 0.62 | 0.912 |
| kerosene | 1.4714 (Tons of standard coal/tons) | 0.616 | 0.906 |
| Diesel fuel | 1.4571 (Tons of standard coal/tons) | 0.657 | 0.957 |
| Fuel oil | 1.4286 (Tons of standard coal/tons) | 0.717 | 1.024 |
| Natural gas | 1.33 (kg standard coal/cubic meter) | 0.523 | 0.696 |
| Electricity | 0.1229 (kg standard coal/kWh) | 1.814 | 0.223 |

Here, we use Equation (6) to calculate the carbon emission intensity coefficient of each industry. Due to the difference between the industry classification in the input–output table and the energy consumption table, the carbon emission coefficient is calculated after aggregation according to the industry characteristics. The results are restored to the 42 industries in the input–output table to obtain the coefficients of the intensity of carbon emissions of each industry in 2012 and 2017 (see Table 3).

**Table 3.** Carbon emission intensity coefficients of various industries in 2012 and 2017.

| Industries | Industry Carbon Emission Intensity Coefficient (Tons/Ten Thousand Yuan) | |
|---|---|---|
| | 2012 | 2017 |
| Agriculture, forestry, animal husbandry and fishery products and services | 0.0517 | 0.0520 |
| Coal Mining Products | 0.6678 | 0.6207 |
| Petroleum and natural gas exploitation products | 0.2507 | 0.2428 |
| Metal mining products | 0.1699 | 0.1646 |
| Nonmetallic ore and other mineral products | 0.2368 | 0.1559 |
| Food and tobacco | 0.0483 | 0.0435 |
| Textiles | 0.1175 | 0.1477 |
| Leather down and products of textile clothing shoes and caps | 0.0336 | 0.0265 |
| Wood products and furniture | 0.0510 | 0.0361 |
| Paper printing and cultural and educational sporting goods | 0.1370 | 0.1168 |
| Petroleum, coking products and nuclear fuel processing products | 1.5541 | 2.1320 |

**Table 3.** *Cont.*

| Industries | Industry Carbon Emission Intensity Coefficient (Tons/Ten Thousand Yuan) | |
| --- | --- | --- |
| | **2012** | **2017** |
| Chemical products | 0.2448 | 0.2609 |
| Non-metallic mineral products | 0.4342 | 0.3466 |
| Metal smelting and rolling products | 0.5826 | 0.7343 |
| Metal products | 0.0833 | 0.0851 |
| General equipment | 0.0583 | 0.0579 |
| Special equipment | 0.0389 | 0.0338 |
| Transportation equipment | 0.0415 | 0.0351 |
| Electrical machinery and equipment | 0.0345 | 0.0303 |
| Communication equipment, computer and other electronic equipment | 0.0295 | 0.0278 |
| Instruments | 0.0389 | 0.0258 |
| Other manufacturing products | 0.4974 | 0.3455 |
| Waste | 0.0175 | 0.0223 |
| Metal products, machinery and equipment repair services | 0.0591 | 0.0343 |
| Production and supply of electricity and heat | 2.1125 | 1.9620 |
| Gas production and supply | 0.2695 | 0.1347 |
| Production and supply of water | 0.4705 | 0.4435 |
| Construction | 0.0182 | 0.0140 |
| Wholesale and retail | 0.0590 | 0.0536 |
| Transport, warehousing and postal services | 0.3270 | 0.2534 |
| Accommodation and catering | 0.0590 | 0.0536 |
| Information transmission, software and information technology services | 0.0379 | 0.0302 |
| Finance | 0.0379 | 0.0302 |
| Real estate | 0.0379 | 0.0302 |
| Lease and business services | 0.0379 | 0.0302 |
| Scientific research and technical services | 0.0379 | 0.0302 |
| Water, environmental and public facilities management | 0.0379 | 0.0302 |
| Resident services, repairs and other services | 0.0379 | 0.0302 |
| Education | 0.0379 | 0.0302 |
| Health and social work | 0.0379 | 0.0302 |
| Culture, sports and entertainment | 0.0379 | 0.0302 |
| Public management, social security and social organizations | 0.0379 | 0.0302 |

According to the results of Table 3, comparing the intensity of carbon emissions of various industries in 2012 and 2017, it can be seen that the intensity of carbon emissions of most industries in China has declined, indicating that between 2012 and 2017, the progress of enterprise management, the improvement of production technology, and the

green development of national policies and environmental regulation have reduced carbon emissions per unit of output in the production process of various industries.

### 3.1.2. Carbon Footprint Analysis of the Financial Industry

In this section, we use model (7) to obtain the carbon footprint of the financial industry in 2012 and 2017. We then further calculated the proportion of carbon emissions in each province to the national carbon emissions and the proportion of carbon emissions of the financial industry in each province to the carbon emissions of the total industries in 2012 and 2017 (see Table 4).

**Table 4.** Carbon emissions and proportions of the financial industry in each province.

| | Financial Sector Carbon Emissions and Share in 2012 | | | Financial Sector Carbon Emissions and Share in 2017 | | |
|---|---|---|---|---|---|---|
| | Financial industry Carbon Emissions Total amount (Tons) | National Financial Industry Carbon Proportion of Emission Provinces | Provincial Carbon Emissions Financial Industry Proportion | Financial Industry Carbon Emissions Total Amount (Tons) | National Financial Industry Carbon Proportion of Emission Provinces | Provincial Carbon Emissions Financial Industry Proportion |
| Beijing | 3,784,808.904 | 9.966% | 0.796% | 2,207,326.88 | 2.474% | 0.976% |
| Tianjin | 960,140.1488 | 2.528% | 0.428% | 1,745,711.507 | 1.956% | 0.749% |
| Hebei | 5,989,566.653 | 15.772% | 1.288% | 3,365,683.833 | 3.772% | 0.691% |
| Shanxi | 393,697.7909 | 1.037% | 0.177% | 520,305.0595 | 0.583% | 0.290% |
| Inner Mongolia | 205,167.9722 | 0.540% | 0.069% | 5,676,095.795 | 6.361% | 3.265% |
| Liaoning | 2,197,562.732 | 5.787% | 0.218% | 2,434,964.484 | 2.729% | 0.847% |
| Jilin | 87,737.09865 | 0.231% | 0.104% | 319,324.5252 | 0.358% | 0.103% |
| Heilongjiang | 448,979.3277 | 1.182% | 0.443% | 2,880,607.352 | 3.228% | 1.415% |
| Shanghai | 2,317,608.275 | 6.103% | 1.552% | 6,464,086.688 | 7.244% | 2.422% |
| Jiangsu | 3,753,693.656 | 9.884% | 0.762% | 11,037,601.76 | 12.370% | 1.423% |
| Zhejiang | 964,834.3764 | 2.541% | 0.334% | 4,078,214.99 | 4.570% | 0.693% |
| Anhui | 1,310,503.484 | 3.451% | 0.613% | 2,140,102.806 | 2.398% | 0.576% |
| Fujian | 277,679.5015 | 0.731% | 0.312% | 2,217,215.633 | 2.485% | 1.156% |
| Jiangxi | 500,585.7538 | 1.318% | 0.430% | 1,940,971.54 | 2.175% | 0.700% |
| Shandong | 3563537.854 | 9.383% | 1.054% | 13,227,151.16 | 14.824% | 1.848% |
| Henan | 2,591,033.314 | 6.823% | 1.709% | 7,200,003.706 | 8.069% | 0.947% |
| Hubei | 9615.475963 | 0.025% | 0.013% | 1,175,464.782 | 1.317% | 0.576% |
| Hunan | 1,016,725.47 | 2.677% | 0.671% | 1,563,788.829 | 1.753% | 0.537% |
| Guangdong | 2,594,581.113 | 6.832% | 0.879% | 4,531,660.85 | 5.079% | 0.561% |
| Guangxi | 658,470.9593 | 1.734% | 0.828% | 2,043,996.417 | 2.291% | 1.079% |
| Hainan | 894,103.1724 | 2.354% | 1.148% | 2,776,333.448 | 3.111% | 3.038% |
| Chongqing | 815671.0643 | 2.148% | 0.799% | 3,466,591.134 | 3.885% | 0.716% |
| Sichuan | 571,603.234 | 1.505% | 0.458% | 1,689,016.481 | 1.893% | 0.653% |
| Guizhou | 214,122.2025 | 0.564% | 0.357% | 864,045.4975 | 0.968% | 0.461% |
| Yunnan | 638,725.5853 | 1.682% | 0.787% | 431,906.5167 | 0.484% | 0.228% |
| Shaanxi | 572,202.5484 | 1.507% | 0.406% | 2,095,639.658 | 2.349% | 0.528% |
| Gansu | 222,662.1954 | 0586% | 0.402% | 511,157.9795 | 0.573% | 0.595% |
| Qinghai | 222,662.1954 | 0.586% | 0.402% | 147,379.6833 | 0.165% | 0.465% |
| Ningxia | 17,246.9545 | 0.045% | 0.054% | 201,358.8254 | 0.226% | 0.292% |
| Xinjiang | 181,194.2143 | 0.477% | 0.252% | 275,705.677 | 0.309% | 0.162% |
| Mean value | 1,265,890.774 | 3.333% | 0.592% | 2,974,313.783 | 3.333% | 0.933% |

As can be seen in Table 4, in terms of the overall carbon footprint of the country, the carbon emissions of the financial industry are small. In 2012 and 2017, the carbon footprint of the financial industry was less than 1% of that of the total industries. Additionally, the proportion in 2017 is slightly higher than that in 2012, which is related to the attributes of the financial industry as a service-oriented industry. Through the analysis of the proportion of carbon emissions from the financial industry in each province to national carbon emissions, it is found that the carbon emissions of the financial industry in Shanghai, Jiangsu, Shandong, Henan, and Guangdong accounted for a relatively high proportion in both 2012 and 2017. Comparatively, the financial industry carbon emissions in Jilin, Qinghai, Ningxia, Hubei, Xinjiang, and Gansu accounted for less than 1% of the national carbon emissions in 2012 and 2017. From the perspective of the change in the proportion of the two years, the proportion of carbon emissions from the financial industry in Beijing, Liaoning, and Hebei decreased significantly in 2017, while that in Inner Mongolia increased significantly. There is no significant change in the proportion of carbon emissions of the financial sector in other provinces. The increase in carbon emissions caused by the growth in economic volume can be effectively controlled through the innovation and optimization of technology and management, to increase the desirable outputs and reduce the undesirable ecological environment output, thus providing a financial guarantee for sustainable development.



The carbon footprint composition of the financial industry and the proportion of upstream and downstream industries in the life cycle of the financial industry are shown in Table 5. From a national perspective, the proportion of direct carbon emissions in the financial industry is relatively small. On the contrary, the proportion of carbon emissions in the upstream and downstream industries of the financial industry is higher than that in the financial industry itself.

**Table 5.** Percentage of carbon emissions from upstream and downstream industries in the financial industry.

| Industry | Upstream | | Downstream | |
|---|---|---|---|---|
| | Industry Share in 2012 | Industry Share in 2017 | Industry Share in 2012 | Industry Share in 2017 |
| Agriculture, forestry, animal husbandry and fishery products and services | 0.0159% | 0.6578% | 0.1483% | 0.3842% |
| Coal Mining Products | 0.0137% | 9.1290% | 4.4027% | 6.0451% |
| Petroleum and natural gas exploitation products | 0.0012% | 1.3520% | 0.5794% | 0.4657% |
| Metal mining products | 0.0000% | 15.6409% | 0.5263% | 0.7031% |
| Nonmetallic ore and other mineral products | 0.0003% | 0.0006% | 0.2479% | 0.2909% |
| Food and tobacco | 0.2822% | 0.1989% | 0.3666% | 0.3741% |
| Textiles | 0.1072% | 0.0109% | 0.6608% | 0.6194% |
| Leather down and products of textile clothing shoes and caps | 0.2967% | 0.1736% | 0.1159% | 0.0775% |
| Wood products and furniture | 0.0973% | 0.0494% | 0.1266% | 0.1000% |
| Paper printing and cultural and educational sporting goods | 11.1070% | 9.3197% | 0.5877% | 0.5790% |
| Petroleum, coking products and nuclear fuel processing products | 14.5921% | 19.7862% | 5.6990% | 14.5911% |
| Chemical products | 0.8437% | 0.3085% | 4.1312% | 5.2008% |
| Non-metallic mineral products | 0.0036% | 0.0113% | 3.4205% | 3.9318% |
| Metal smelting and rolling products | 0.0334% | 0.0001% | 10.4864% | 16.8528% |
| Metal products | 0.1644% | 0.0160% | 0.3530% | 0.4874% |
| General equipment | 0.3734% | 0.0963% | 0.3022% | 0.2929% |
| Special equipment | 0.1727% | 0.1449% | 0.1796% | 0.1905% |
| Transportation equipment | 0.1153% | 0.0748% | 0.2764% | 0.2713% |
| Electrical machinery and equipment | 0.0139% | 0.0107% | 0.1899% | 0.1847% |
| Communication equipment, computer and other electronic equipment | 0.1734% | 0.1553% | 0.1386% | 0.2121% |
| Instruments | 0.0016% | 0.0008% | 0.0432% | 0.0513% |
| Other manufacturing products | 0.0910% | 0.0240% | 0.1587% | 0.4092% |
| Waste waste | 0.0000% | 0.0227% | 0.0053% | 0.0074% |
| Metal products, machinery and equipment repair services | 0.0392% | 0.6569% | 0.0164% | 0.7741% |
| Production and supply of electricity and heat | 41.5493% | 0.1276% | 47.7965% | 5.1242% |
| Gas production and supply | 0.0214% | 0.2306% | 0.1735% | 0.1465% |
| Production and supply of water | 0.9795% | 3.2132% | 0.5456% | 27.2436% |
| Construction | 0.1276% | 0.2618% | 0.3653% | 1.0430% |
| Wholesale and retail | 0.7148% | 1.8273% | 2.2909% | 3.7389% |
| Transport, warehousing and postal services | 10.5698% | 15.3055% | 11.7113% | 2.1541% |
| Accommodation and catering | 4.1746% | 3.3190% | 0.2628% | 0.5378% |
| Information transmission, software and information technology services | 2.1747% | 6.4639% | 0.1610% | 2.4760% |
| Finance | 3.5543% | 3.7173% | 1.0634% | 1.5116% |
| Real estate | 2.1669% | 5.9927% | 0.8292% | 1.7190% |
| Lease and business services | 4.0472% | 0.0000% | 0.8901% | 0.0670% |
| Scientific research and technical services | 0.0549% | 0.1586% | 0.1608% | 0.2608% |

**Table 5.** *Cont.*

| Industry | Upstream | | Downstream | |
|---|---|---|---|---|
| | Industry Share in 2012 | Industry Share in 2017 | Industry Share in 2012 | Industry Share in 2017 |
| Water, environmental and public facilities management | 0.0705% | 0.2410% | 0.0632% | 0.1115% |
| Resident services, repairs and other services | 0.2853% | 0.3307% | 0.1314% | 0.1792% |
| Education | 0.4695% | 0.5654% | 0.1169% | 0.1816% |
| Health and social work | 0.0091% | 0.0119% | 0.0380% | 0.0550% |
| Culture, sports and entertainment | 0.3880% | 0.3079% | 0.0684% | 0.1101% |
| Public management, social security and social organizations | 0.1035% | 0.0843% | 0.1690% | 0.2440% |
| Total carbon emissions (10,000 tons) | 3487.80 | 11658.15 | 4665.17 | 12178.92 |

The primary sources of carbon emissions of upstream enterprises of the financial industry are transportation warehousing and postal, paper printing and cultural and educational sports goods, accommodation and catering, finance, real estate, and software and information service technology. In 2012, electricity and heat production and supply, as well as leasing and business services, contributed more carbon emissions. In 2017, metal mining and coal mining products contributed more carbon emissions.

The primary sources of carbon emissions of downstream enterprises of the financial industry are transportation warehousing and postal, metal refining and rolling processing products, power and heat production and supply, chemical production, non-metallic mineral products, the coal mining and dressing industry, the wholesale and retail industry, petroleum coking products, and nuclear fuel processing. In 2012, the financial industry provided more intermediate inputs for the production and supply of water and information transmission, software, and information technology services, resulting in more carbon emissions. In 2017, the industries with more carbon emissions were concentrated in the downstream industries of the financial industry. Almost 70% of carbon emissions come from transportation, warehousing and postal services, metal refining and rolling processing products, and the production and supply of electricity and heat.

On the whole, the carbon footprint of the financial industry in providing services to the downstream industry is much larger than that of the upstream industry. The carbon footprint provided by different industries to the financial industry varies greatly, which also reflects that the industrial structure has an important impact on the ecological efficiency of the financial industry. Financial institutions should formulate a reasonable capital allocation structure, optimize capital support objects, provide sufficient financial support to high-emission enterprises or fields with green transformation, and guide the green transformation of industries.

In Figure 2, we present the proportion of the final demand composition of the regional financial industry. From the final demand, the financial industry's carbon footprint is caused by four categories of final demand: Rural consumption, urban demand, government consumption, export, or interprovincial outflow. Among them, government consumption accounted for a small proportion, 3.45% in 2012 and 1.99% in 2017. Urban residents' demand accounted for 53.95% in 2012 and 50.48% in 2017. In 2012, the proportion of domestic inter-provincial flows accounted for 29.79%, and in 2017, the proportion of domestic inter-provincial flows accounted for 31.17%. In 2012, the rural residents' demand accounted for 12.81%, and in 2017, the rural residents' demand accounted for 16.37%, which reflects the process of the financial industry's development from urban to rural areas. In summary, the financial industry's carbon footprint is mainly derived from consumer consumption, followed by inter-provincial flows. The external demand of the financial industry in the four municipalities accounts for a large proportion, indicating that it plays a role as a financial hub. Except for the four municipalities directly under the Central

Government, Beijing, Tianjin, Shanghai, and Chongqing, the demand for urban residents in most provinces is relatively large (see Figure 2).

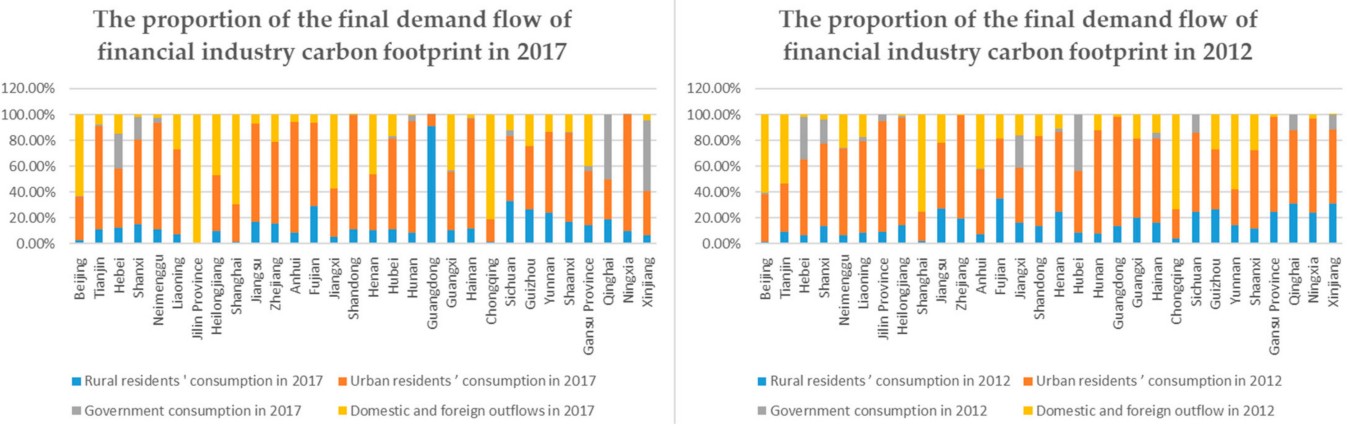

**Figure 2.** Proportion of final demand composition of the regional financial industry.

### 3.2. Calculation and Analysis of Ecological Efficiency

3.2.1. Descriptive Statistics of Input–Output Indicators

In this paper, the employment scale and fixed asset investment data of the financial industry are from the statistical yearbooks of each province in china in 2012 and 2017. The value-added data at the end of the year are from the National Bureau of Statistics. The carbon footprint data of the financial industry in each province are calculated above. The observed values of each indicator are shown in Table 6. Table 6 shows that the investment indicators of the financial industry are increasing, and the value-added and carbon footprint of the financial industry are also increasing.

**Table 6.** Descriptive statistics of input–output indicators.

| | Year | Minimum | Maximum | Mean Value | Standard Deviation |
|---|---|---|---|---|---|
| Employed persons (ten thousand people) | 2012 | 0.80 | 57.70 | 18.53 | 13.11 |
| | 2017 | 2.29 | 64.10 | 23.39 | 17.58 |
| Investment in fixed assets (100 million yuan) | 2012 | 0.69 | 97.85 | 30.98 | 27.12 |
| | 2017 | 0.19 | 136.89 | 35.05 | 34.48 |
| Value-added (100 million yuan) | 2012 | 83.73 | 3171.96 | 1014.57 | 911.38 |
| | 2017 | 256.46 | 6850.70 | 2146.45 | 1783.95 |
| Carbon emissions (ten thousand tons) | 2012 | 0.96 | 375.37 | 104.73 | 111.67 |
| | 2017 | 14.74 | 1322.72 | 297.43 | 309.51 |

3.2.2. Ecological Efficiency Calculation

Using model (8) and formula (9), we calculated the ecological efficiency of the financial efficiency of 30 provinces under natural disposability and managerial disposability in 2012 and 2017. The specific scores are reported in Table 7.

In 2012, it was observed that under natural disposability, managers pursued earnings performance. Ecological efficiency scores of the financial industry in Beijing, Shanxi, Jilin, Shanghai, Jiangsu, Zhejiang, Fujian, Hubei, Guangdong, Chongqing, Qinghai, and Ningxia are 1, which constitute the production frontier. The financial industry eco-efficiency scores of Tianjin, Henan, Hainan, Guizhou, and Yunnan range from 0.9 to 1, which are relatively close to the frontier. The ecological efficiency scores of the financial industry of Sichuan and Xinjiang are between 0.8 and 0.9. The ecological efficiency scores of the financial industry in Inner Mongolia and Shandong are even lower, between 0.6 and 0.8. At the

same time, the scores of ecological efficiency of the financial industry in Hebei, Liaoning, Heilongjiang, Anhui, Jiangxi, Hunan, Guangxi, Shaanxi, and Gansu are all less than 0.6, which are the regions with poor performance. Differences in the ecological efficiency scores of the financial industry in different provinces can reflect differences in regional economic development. Specifically, Beijing, Shanghai, Jiangsu, Guangdong, Chongqing, Zhejiang, Fujian, Hubei, and other developed areas have higher ecological efficiency scores of the financial industry. The financial industry development endowments in Qinghai, Ningxia, Jilin, and Shanxi are not high.

**Table 7.** Eco-efficiency under natural disposability and management disposability.

| Area | Province | EN | | EM | |
|---|---|---|---|---|---|
| | | **2012** | **2017** | **2012** | **2017** |
| Eastern | Beijing | 1.000 | 1.000 | 0.828 | 1.000 |
| | Fujian | 1.000 | 0.877 | 0.739 | 0.590 |
| | Guangdong | 1.000 | 1.000 | 1.000 | 1.000 |
| | Hainan | 0.908 | 1.000 | 0.044 | 0.092 |
| | Hebei | 0.327 | 0.437 | 0.163 | 0.818 |
| | Jiangsu | 1.000 | 1.000 | 1.000 | 1.000 |
| | Liaoning | 0.383 | 1.000 | 0.182 | 0.554 |
| | Shandong | 0.696 | 0.299 | 0.479 | 1.000 |
| | Shanghai | 1.000 | 1.000 | 0.790 | 0.715 |
| | Tianjin | 0.912 | 0.673 | 0.511 | 0.666 |
| | Zhejiang | 1.000 | 0.611 | 1.000 | 0.832 |
| | Eastern average | 0.839 | 0.809 | 0.612 | 0.752 |
| Central | Anhui | 0.148 | 0.510 | 0.262 | 0.596 |
| | Henan | 0.945 | 0.383 | 0.178 | 0.286 |
| | Heilongjiang | 0.214 | 0.244 | 0.250 | 0.293 |
| | Hubei | 1.000 | 1.000 | 1.000 | 1.000 |
| | Hunan | 0.505 | 1.000 | 0.372 | 1.000 |
| | Jilin | 1.000 | 0.919 | 0.081 | 1.000 |
| | Jiangxi | 0.234 | 0.407 | 0.013 | 0.465 |
| | Shanxi | 1.000 | 1.000 | 0.904 | 1.000 |
| | Central average | 0.631 | 0.683 | 0.383 | 0.705 |
| Western | Gansu | 0.067 | 0.597 | 0.032 | 0.603 |
| | Guangxi | 0.256 | 0.693 | 0.133 | 0.434 |
| | Guizhou | 0.880 | 0.496 | 0.031 | 0.509 |
| | Inner Mongolia | 0.699 | 0.284 | 0.196 | 0.152 |
| | Ningxia | 1.000 | 0.878 | 0.438 | 0.926 |
| | Qinghai | 1.000 | 1.000 | 0.346 | 1.000 |
| | Shaanxi | 0.307 | 0.418 | 0.118 | 0.427 |
| | Sichuan | 0.793 | 0.951 | 0.695 | 0.961 |
| | Xinjiang | 0.806 | 0.985 | 0.035 | 1.000 |
| | Yunnan | 0.919 | 1.000 | 0.110 | 1.000 |
| | Chongqing | 1.000 | 0.967 | 0.369 | 0.383 |
| | Western average | 0.703 | 0.752 | 0.228 | 0.672 |

Under managerial disposability, we should not only improve the desirable output, but also reduce the undesirable output, that is, environmental performance is also essential. The results show that the scores of ecological efficiency of the financial industry in Jiangxi, Zhejiang, Hubei, and Guangdong are all 1, which are effective units. The scores in Shanxi, Beijing, Shanghai, and Fujian are between 0.6 and 0.8; it shows that most provinces have low efficiency scores and the development of environmental awareness and progress in environmental protection technology is weak. At the same time, it can be found that the interprovincial differences in ecological efficiency are apparent. Therefore, it is further divided into regions. There is much literature that divides China into East, Middle, and West regions, and then analyzes interprovincial problems. Based on the existing divi-

sion standards of eastern, central, and western areas, comparative analysis shows that the ecological efficiencies of the financial industry in the three regions are significantly different [35]. The ecological efficiency of the financial industry in the eastern region is better than that of the western and central regions, and the western region is slightly better than the central region.

In Figure 3, we compare the ecological efficiency scores of the financial industry in natural disposability and managerial disposability in 2012 and 2017.

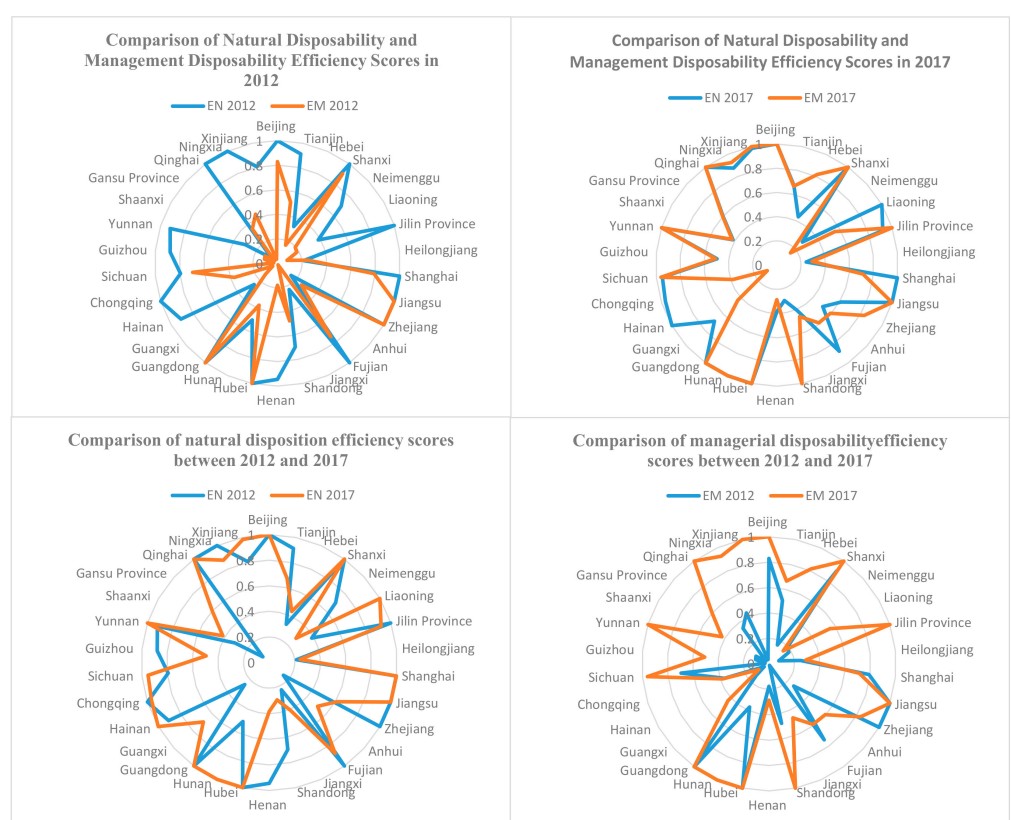

**Figure 3.** Eco-efficiency scores of the financial industry under EN and EM in 2012 and 2017.

The comparative analysis of efficiency scores in the same year shows that in 2012, most provinces had a higher ecological efficiency score under natural disposability, and more than 60% of provinces scored more than 0.8. Under managerial disposability, the efficiency score of more than half of the provinces is less than 0.5. It reflects that most provinces ignored environmental protection and did not pay attention to the improvement of environmental protection technology and management level before 2012. Overall, for Guangdong, Hubei, Zhejiang, and Jiangsu, under both types of disposability, the ecological efficiency score of the financial industry is 1. Beijing, Shanxi, Shanghai, and Fujian scored 1 under natural disposability and scored higher under managerial disposability. Sichuan has high efficiency scores under both types of disposability with little difference.

The ecological efficiency of the financial industry in the above provinces is significantly better than in other provinces in 2012. There are two reasons. First, the regional economy with unique advantages in geographical location, resource endowment, and energy structure is relatively developed, and the financial industry's ecological efficiency score is higher. Second, when the government implements environmental policies, it will choose economically developed regions as pilots to promote industrial technological innovation and guide the capital flow to ecological protection enterprises. In contrast, Qinghai, Ningxia, Jilin, and Chongqing scored 1 for natural disposal and scored lower for managerial disposal. Compared to 2012, 2017 had a relatively balanced efficiency score under the two types of disposal.

Comparing different years under the same disposal, it is observed that the efficiency scores of each province in 2017 under the two types of disposal are significantly better than those in 2012. It can be explained by two factors: First, the green financial system is included in the top-level design of national development. At the end of 2012, the 18th National Congress of the Communist Party of China put forward requirements for constructing ecological civilization. In 2015, the State Council proposed the 'overall plan for the reform of the ecological civilization system', which has played a role in promoting the improvement of green production technology in various industries and the ecological efficiency of the financial industry. Second, from 2012 to 2017, China's financial industry developed rapidly. Under the guidance of the concept of green innovation, the green development of the financial industry can promote technological innovation to curb carbon emissions and increase eco-efficiency scores. It shows that in the process of economic development, regions should pay attention to the role of the financial industry in guiding enterprises to innovate in energy conservation and emission reduction and low green carbon.

Combining Table 7 and Figure 3 and the main source of the carbon footprint of the financial industry in Chapter 2 (Table 5), it is concluded that the interprovincial differences in the ecological efficiency of the financial industry are very large, mainly due to regional industrial structure differences. Provinces with a relatively developed tertiary industry have a relatively high ecological efficiency of the financial industry. By analyzing and comparing the source of the carbon footprint of the financial industry in each province in 2012 and 2017 (Table 5), it is found that the provinces with higher ecological efficiency of the financial industry provide more funds and services to industries with low carbon emissions, such as Beijing, Shanghai, Jiangsu, Guangdong, and other provinces. The tertiary industry is relatively developed, and the flow of funds in the tertiary industry is relatively large. In provinces where the secondary industry is relatively developed, the financial industry funds and services flow more to industries with higher carbon emissions, resulting in lower ecological efficiency of the financial industry, such as Liaoning, Shandong, and other heavy industrial provinces. At the same time, in the provinces with a low endowment of financial industry development, the economy is relatively underdeveloped and there is no high carbon footprint. It can be seen that the regional industrial structure has a significant impact on the ecological efficiency of the financial industry. The improvement of the ecological efficiency of the financial industry needs to further optimize its resource allocation and flow resources to green energy-saving enterprises.

### 3.3. Emission Reduction and Value-Added Potential Analysis

From the perspective of the ecological efficiency score, most provinces have not reached an efficiency score of 1, that is, they are not effective. It indicates that there is room to improve the ecological efficiency in inefficient provinces. The DEA model can provide an improvement direction for them. Through efficiency scores and the value of slack variables, the carbon emission reduction potential and the value-added potential of the financial industry in each province can be calculated (see Table 8).

**Table 8.** Regional emission reduction and value-added potential.

| Disposition | Natural Disposition | | | | Managerial Disposability | | | |
|---|---|---|---|---|---|---|---|---|
| | Value-Added Potential | | Carbon Emission Reduction Potential | | Value-Added Potential | | Carbon Emission Reduction Potential | |
| | 2012 | 2017 | 2012 | 2017 | 2012 | 2017 | 2012 | 2017 |
| Beijing | 0.00 | 0.00 | 0.00 | 0.00 | 436.50 | 0.00 | 65.12 | 0.00 |
| Tianjin | 88.46 | 637.72 | 8.48 | 57.04 | 490.20 | 651.48 | 46.99 | 58.27 |
| Hebei | 614.58 | 1156.36 | 402.89 | 189.63 | 764.73 | 372.70 | 501.32 | 61.12 |
| Shanxi | 0.00 | 0.00 | 0.00 | 0.00 | 61.39 | 0.00 | 3.78 | 0.00 |
| Inner Mongolia | 150.99 | 787.98 | 6.17 | 406.66 | 403.83 | 932.69 | 16.50 | 481.34 |
| Liaoning | 597.77 | 0.00 | 236.90 | 0.00 | 793.19 | 917.94 | 314.34 | 108.56 |
| Jilin | 0.00 | 55.25 | 0.00 | 2.58 | 224.79 | 0.00 | 14.58 | 0.00 |
| Heilongjiang | 381.22 | 741.39 | 84.33 | 217.75 | 363.61 | 693.85 | 80.43 | 203.78 |
| Shanghai | 0.00 | 0.00 | 0.00 | 0.00 | 514.44 | 1520.16 | 81.47 | 184.34 |

**Table 8.** *Cont.*

| Disposition | Natural Disposition | | | | Managerial Disposability | | | |
|---|---|---|---|---|---|---|---|---|
| | Value-Added Potential | | Carbon Emission Reduction Potential | | Value-Added Potential | | Carbon Emission Reduction Potential | |
| | 2012 | 2017 | 2012 | 2017 | 2012 | 2017 | 2012 | 2017 |
| Jiangsu | 0.00 | 0.00 | 0.00 | 0.00 | 0.00 | 0.00 | 0.00 | 0.00 |
| Zhejiang | 0.00 | 1373.56 | 0.00 | 158.55 | 0.00 | 592.72 | 0.00 | 68.42 |
| Anhui | 526.18 | 814.79 | 191.24 | 104.82 | 456.08 | 672.74 | 165.76 | 86.54 |
| Fujian | 0.00 | 251.81 | 0.00 | 27.16 | 265.42 | 842.42 | 11.61 | 90.87 |
| Jiangxi | 316.47 | 704.00 | 76.22 | 115.09 | 407.73 | 635.41 | 98.20 | 103.87 |
| Shandong | 588.30 | 2597.59 | 184.70 | 926.80 | 1009.05 | 0.00 | 316.79 | 0.00 |
| Henan | 55.46 | 1560.37 | 23.46 | 444.00 | 833.21 | 1807.20 | 352.40 | 514.24 |
| Hubei | 0.00 | 0.00 | 0.00 | 0.00 | 0.00 | 0.00 | 0.00 | 0.00 |
| Hunan | 287.07 | 0.00 | 100.73 | 0.00 | 363.94 | 0.00 | 127.70 | 0.00 |
| Guangdong | 0.00 | 0.00 | 0.00 | 0.00 | 0.00 | 0.00 | 0.00 | 0.00 |
| Guangxi | 426.09 | 390.80 | 93.05 | 62.73 | 496.94 | 720.63 | 108.52 | 115.67 |
| Hainan | 12.05 | 0.00 | 2.68 | 0.00 | 124.89 | 289.09 | 27.73 | 252.22 |
| Chongqing | 0.00 | 60.35 | 0.00 | 11.53 | 577.92 | 1119.53 | 84.12 | 213.98 |
| Sichuan | 269.35 | 162.01 | 19.74 | 8.28 | 397.57 | 128.46 | 29.14 | 6.57 |
| Guizhou | 43.92 | 396.81 | 4.95 | 43.52 | 354.46 | 386.58 | 39.97 | 42.40 |
| Yunnan | 43.69 | 0.00 | 9.57 | 0.00 | 481.77 | 0.00 | 105.49 | 0.00 |
| Shaanxi | 381.88 | 756.41 | 82.69 | 121.93 | 486.05 | 744.86 | 105.25 | 120.07 |
| Gansu | 172.08 | 223.11 | 37.98 | 20.60 | 178.60 | 220.02 | 39.41 | 20.32 |
| Qinghai | 0.00 | 0.00 | 0.00 | 0.00 | 54.73 | 0.00 | 2.43 | 0.00 |
| Ningxia | 0.00 | 38.51 | 0.00 | 2.46 | 94.05 | 23.24 | 1.65 | 1.49 |
| Xinjiang | 69.93 | 10.04 | 7.13 | 0.42 | 347.78 | 0.00 | 35.45 | 0.00 |
| Total | 5025.49 | 12718.86 | 1572.89 | 2921.56 | 10982.87 | 13271.73 | 2776.17 | 2734.06 |

Table 8 shows the value-added potential and the emission reduction potential of the financial industry in each province in 2012 and 2017. By comparison, we found that the financial services and funds of Shandong, Hebei, Liaoning, Anhui, Heilongjiang, Henan, Hunan, Shaanxi, Inner Mongolia, and other provinces flowed more to industry and manufacturing, resulting in higher carbon emissions and greater carbon emission reduction potential of the financial industry.

Through data analysis, we can determine that the carbon emission reduction potential of the financial industry under natural disposability in 2012 is 15.72 million tons, and the value-added potential of the financial industry is 50.25 billion yuan. The carbon emission reduction potential of the financial industry under management disposability is 27.76 million tons, and the value-added potential of the financial industry is 109.82 billion yuan. In 2017, the carbon emission reduction potential of the financial industry under natural disposability is 29.21 million tons, and the value-added potential of the financial industry is 127.18 billion yuan. The carbon emission reduction potential of the financial industry under managerial disposability is 27.34 million tons, and the value-added potential of the financial industry is 132.71 billion yuan. The reduction in emissions and value-added potential under natural disposability is less than that under managerial disposability. Management disposability refers to management change and technological innovation, which indicates that the study of eco-efficiency under management disposability can guide emission reductions more scientifically, achieve higher emission reduction targets, and obtain higher added value. It can be seen that the financial industry can provide green financial security for the whole society to cope with carbon emissions and achieve effective added value by optimizing capital allocation, strengthening the capital management system, and increasing the green flow of its service products. In 2017, there were significant improvements in technology and management compared to 2012, but there were still provinces where the financial sector was less eco-efficient.

## 4. Conclusions and Policy Suggestions

Based on life cycle theory, in this paper, we used the input–output analysis method to construct a carbon footprint model of the financial industry. By using this model, we calculated the carbon emissions generated for the financial industry's own development, and the carbon emissions generated by other industries receiving financial services were calculated. Then the carbon footprint of the entire life cycle of the financial industry was calculated. The carbon footprint of the financial industry is used as an undesirable output,

and we then constructed the ecological efficiency evaluation index system of the financial industry of 30 provinces in China. Finally, we applied the SRAM-DEA model to evaluate the ecological efficiency of the financial industry in China in 2012 and 2017 under natural and managerial disposability, respectively.

The empirical results show that (1) the intensity of carbon emissions of most industries in China declined between 2012 and 2017. (2) High energy consumption and high pollution industries are the main sources of carbon emissions in the financial industry. (3) The industrial structure has a significant impact on the ecological efficiency of the financial industry. (4) There are significant differences in the reduction of emissions and the potential for added value in the financial industry under natural and managerial disposal. In general, the potential for disposal under the managerial type is greater.

Paying attention to the protection of the ecological environment is a necessary condition for China's high-quality economic development. As the core industry of economic development, the financial industry must play a guiding role in economic development. Government departments should strengthen environmental protection supervision of the financial industry and other industries to improve the ecological efficiency of the financial industry, realize the green cycle of funds, and promote social sustainable development. Based on the above research, the following policy recommendations are given.

The first recommendation is for financial institutions: (1) To optimize the capital allocation structure, financial institutions should fully consider the environmental performance of companies when choosing their cooperative companies. The green transformation of enterprises should be guided by building a scientific corporate environmental evaluation index system and optimizing the funds and services invested in high-pollution, high-energy-consuming enterprises. (2) Financial institutions themselves should develop greenly. Specifically, they should replace the traditional office mode with a paperless office and an online office. In this way, office efficiency can be improved, and meanwhile, carbon emissions can be reduced effectively in their own development process. (3) Financial institutions must increase the necessary financial support for companies in transition and emerging green industries to solve the problem of insufficient green investment and provide greater impetus for green economic development. (4) Financial institutions should strengthen capital supervision, accurately assess the environmental benefits of capital service objects in the production process, and prevent the risk of ecological environmental pollution.

The second recommendation is for government agencies: (1) Government agencies are advised to give full play to government regulatory functions, formulate pollutant emission indicators, limit the development of high-pollution industries, promote technological innovation and management reform, and ultimately reduce carbon emissions from the root. (2) They should support the development of emerging green industries, provide a policy guarantee for the development of green industries, promote the transformation of social supply to green, and also promote sustainable economic development. (3) They should work closely with financial institutions to establish a green project library to address information asymmetry in capital allocation. (4) According to local economic development, they should formulate policies to promote the innovation of production technology, encourage enterprises to upgrade the industrial structure, improve the regional emission reduction ability and value-added ability, and ultimately realize the coordinated development of the economy and the environment.

**Author Contributions:** Conceptualization, X.C., K.W., G.W., Y.L., W.L., W.S. and J.S.; methodology, K.W. and G.W.; software, K.W.; validation, K.W.; formal analysis, K.W.; investigation, X.C., K.W. and G.W.; resources, X.C., W.S. and W.L.; data curation, K.W.; writing—original draft preparation, X.C. and K.W.; writing—review and editing, X.C., K.W., Y.L. and J.S.; visualization, K.W.; supervision, X.C., W.S. and W.L.; project administration, X.C. and W.S.; funding acquisition, X.C. and W.S. All authors have read and agreed to the published version of the manuscript.

**Funding:** This research was funded by the National Natural Science Foundation of China (NSFC), grant number, 11971259, and the National Social Science Found of China (NSSFC), grant number, 21BTJ072.

**Institutional Review Board Statement:** No applicable.

**Informed Consent Statement:** No applicable.

**Data Availability Statement:** The data are not publicly available due to privacy restrictions.

**Acknowledgments:** This study was sponsored by the National Natural Science Foundation of China (NSFC: 11971259), and the National Social Science Found of China (NSSFC: 21BTJ072), which is highly acknowledged.

**Conflicts of Interest:** The authors declare no conflict of interest.

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
