# Peer review of "Evaluation and Empirical Research on Eco-Efficiency of Financial Industry Based on Carbon Footprint in China"

_sustainability, doi:10.3390/su142013677_

Round 1
Reviewer 1 Report (Previous Reviewer 2)
1-I suggest the authors revise the introduction of the study per the comments raised. Authors can also use the following points below as a guideline to help them come out with an interesting introduction that is more scientific.
* Background & Significance: (What general background does the reader need in order to understand the manuscript and how important is it in the context of scientific research).
* Problem definition: (What are the research questions to fill in the gaps of the existing knowledge body or methodology).
* Motivations & Objectives: (Why are you conducting the study and what are the goals to achieve?)
Author Response
Please see the attachment.

Reviewer 2 Report (Previous Reviewer 3)
The authors have improved the manuscript according to the suggestions of reviewers. I am fine with the current manuscript.
Author Response
Please see the attachment.

This manuscript is a resubmission of an earlier submission. The following is a list of the peer review reports and author responses from that submission.
Round 1
Reviewer 1 Report
1. In the Literature Review Section, this manuscript lacks some discussion about the progress of research on eco-efficiency in the financial industry. The research structure for each section should also be proposed.
2. The authors take the measurement of carbon footprint and eco-efficiency in the financial industry as an innovation point, however this innovation is not outstanding enough. Some theoretical analysis is needed to clearly explain the application of life cycle theory, the particularity of the financial industry, and the possible ecological problems.
3. The definition of financial industry ecological efficiency is not clear. The financial industry ecological efficiency is defined as the ratio of added value of financial products and services and environmental impact. In fact, most of the relevant studies consider ecological efficiency as the total factor input and output of the ecological environment problems. However, the analysis model in this paper is also treated added value of financial products and services and the environmental impact as expected output and the unexpected output.
4. The authors try to use carbon footprint as a key indicator to measure environmental factors, but the issue of eco-efficiency is not only related to carbon emissions, it should also include other undesired environmental pollutants. The study needs to further describe the ecological problems of the financial industry through theoretical analysis.
5. In terms of data acquisition, the current provincial database on carbon emissions has been updated at least to 2020. Since the "dual carbon" goal was put forward, the data of various industries have changed significantly. On the one hand, carbon intensity data after 2017 should be supplemented to illustrate the latest research results. On the other hand, it should be explained whether the output value data has been treated at constant price.
6. The authors need to make further in-depth analysis of research results in Discussion Section. For example, by comparing the high and low carbon footprints, how would the authors conclude that financial institutions should formulate reasonable policies to reduce funds and services for high-emitting industries or fields. Take the example of iron and steel industry, it has high carbon emissions due to problems with its production process, the carbon footprint must be high. However, judging from the current policy requirements for low-carbon emission reduction and carbon market transactions, it is essential to have sufficient financial service guarantees to support its transformation.
7. A logical check and stylistic revision of the English of the thesis is still necessary.

Reviewer 2 Report
- I suggest the authors revise the introduction of the study per the comments raised. Authors can also use the following points below as a guideline to help them come out with an interesting introduction that is more scientific.
• Background & Significance: (What general background does the reader need in order to understand the manuscript and how important is it in the context of scientific research).
• Problem definition: (What are the research questions to fill in the gaps of the existing knowledge body or methodology).
• Motivations & Objectives: (Why are you conducting the study and what are goals to achieve?) - Policy implications are not strongly related to the research findings, so it needs to be improved.
- English writing needs to be improved, for example, many typos in the manuscript.
- Revised all manuscript according to the "author guideline of sustainability journal".
Reviewer 3 Report
The whole study is well designed and clearly presented, and I think the results are robust. However, my only suggestion is that the conclusion part is too long in current manuscript. Authors should summarize the main results of empirical analysis in one concise paragraph, instead of 4 paragraphs which contain all detailed findings. Those detailed findings can be presented in part 3.